# B Vitamins, Glucoronolactone and the Immune System: Bioavailability, Doses and Efficiency

**DOI:** 10.3390/nu16010024

**Published:** 2023-12-20

**Authors:** Camelia Munteanu, Betty Schwartz

**Affiliations:** 1Department of Plant Culture, Faculty of Agriculture, University of Agricultural Sciences and Veterinary Medicine, 400372 Cluj-Napoca, Romania; 2The Institute of Biochemistry, Food Science and Nutrition, The School of Nutritional Sciences, Robert H. Smith Faculty of Agriculture, Food and Environment, The Hebrew University of Jerusalem, Rehovot 76100, Israel

**Keywords:** B vitamins, riboflavin, niacin, cobalamin, glucuronolactone, immune system, bioavailability

## Abstract

The present review deals with two main ingredients of energy/power drinks: B vitamins and glucuronolactone and their possible effect on the immune system. There is a strong relationship between the recommended daily dose of selected B vitamins and a functional immune system. Regarding specific B vitamins: (1) Riboflavin is necessary for the optimization of reactive oxygen species (ROS) in the fight against bacterial infections caused by *Staphylococcus aureus* and *Listeria monocytogenes*. (2) Niacin administered within normal doses to obese rats can change the phenotype of skeletal fibers, and thereby affect muscle metabolism. This metabolic phenotype induced by niacin treatment is also confirmed by stimulation of the expression of genes involved in the metabolism of free fatty acids (FFAs) and oxidative phosphorylation at this level. (3) Vitamin B5 effects depend primarily on the dose, thus large doses can cause diarrhea or functional disorders of the digestive tract whereas normal levels are effective in wound healing, liver detoxification, and joint health support. (4) High vitamin B6 concentrations (>2000 mg per day) have been shown to exert a significant negative impact on the dorsal root ganglia. Whereas, at doses of approximately 70 ng/mL, sensory symptoms were reported in 80% of cases. (5) Chronic increases in vitamin B12 have been associated with the increased incidence of solid cancers. Additionally, glucuronolactone, whose effects are not well known, represents a controversial compound. (6) Supplementing with D-glucarates, such as glucuronolactone, may help the body’s natural defense system function better to inhibit different tumor promoters and carcinogens and their consequences. Cumulatively, the present review aims to evaluate the relationship between the selected B vitamins group, glucuronolactone, and the immune system and their associations to bioavailability, doses, and efficiency.

## 1. Introduction

There is a strong relationship between selected B vitamins and the immune system, especially when over-administered since their amount in different processed food products and supplements exceeds the recommended daily dose. At the same time, glucuronolactone, whose effects are not well understood, exemplifies a controversial compound. Effects of B vitamins and glucoronolactone gained popularity since these substances are in large quantities in different supplements and energy drinks (ED). Glucoronolactone as well as B complex supplements are widely consumed, due to their producer’s promises to boost cognitive and physical performance. However, these manufacturers do not provide any real information regarding overdoses and long-term consequences [1]. Even children and adolescents nowadays consume ED and even ED with alcohol (EDwA) on a regular basis. The daily dose recommendations differ significantly as a function of age, however clear recommendations regarding their safety are lacking [2].

In addition, the mixed consumption of EDwA has become a common trend at parties [3]. Energy beverages contain caffeine and taurine, the main compounds, vitamins from the B group, glucuronolactone, and additionally, certain extracts. The acute administration of energy drinks in moderate doses seems to increase sports performance [4]. In the case of excessive or long-term administration the effects are controversial, because of the health concern associated with these beverages [5]. In most cases, the adverse effects are attributed to caffeine and taurine. Nonetheless, not enough attention has been directed towards the effects of the number and amounts of B vitamins (B2, B3, B5, B6 and B12), which are cofactors for different metabolic reactions [6]. Excessing consumption of B vitamins alone or together with other compounds can cause health consequences [7]. Moreover, when they are consumed in the specific combination observed in ED they can exert synergistic and even complementary negative effects [8]. In such a manner, even their absorption may be affected and can be increased or decreased compared to the basic situation [9]. Another controversial compound is glucuronolactone, whose effects remain quite unknown. In recent years, more studies have revealed the toxic properties of high doses of it [10].

In regard to the effects of B vitamins, catabolic and anabolic metabolism depend on B vitamins, regarded also as B-complex vitamins. The names of the B vitamins are: cobalamin (B12), pyridoxine (B6), biotin (B7), thiamine (B1), riboflavin (B2), niacin (B3), pantothenic acid (B5), and biotin (B6). Organisms require these eight water-soluble vitamins, which are eliminated in urine when consumed at high doses, as they are regarded as vital for well-being. The major physiological activities of cells, including the brain and nervous system, are supported by the B vitamins as coenzymes in a number of enzymatic processes [11]. The electron transport chain and citric acid cycle, the mitochondrial metabolism of amino acids, glucose, and fatty acids all can be severely and adversely affected by any B vitamin shortage (Table 1).

The following are the most common reported effects of overdoses of the selected vitamins: (1) Pharmacologic riboflavin (RF) overdoses that are taken regularly may react with light. The effects can produce failures in the cells. Thus, free RF becomes a potent oxidant due to the photoreactive properties of the isoalloxazine ring, which can generate damaging peroxides or other reactive oxygen species, as well as generate an unusual tryptophan metabolite [18]. (2) Niacin is presented in a larger quantity than the recommended daily dose. Considering that it is involved in many molecular reactions, high doses can potentiate these pathways [19]. Vitamin B3 (VB3) has long been used to decrease the lipid profile. Moreover, it is widely used by people as a supplement, without adequate knowledge of the repercussions caused by large doses of this vitamin [20]. (3) Vitamin B5 (VB5), like vitamin B3 (VB3), is also administrated in patients with hyperlipidemia, due to its involvement in the synthesis of triglycerides and the metabolism of lipoproteins [21]. Regarding this effect, the research had contradictory results. It has been shown that only the VB5 administration in the form of pantethine declined the blood lipid concentrations [22], while pantothenic acid individually does not exert a similar effect [23]. (4) In the case of vitamin B6 (VB6), it has been observed that its plasma concentration is usually increased after multivitamin supplementation. However, the highest blood levels of VB6 have been reported after biliopancreatic diversion (BPD) due to the daily consumption of energy drinks [24]. (5) Vitamin B12 (VB12), like the other vitamins, is also provided in an amount above the daily dose. Generally, high doses of VB12 are not considered toxic because the body does not store the excess of this vitamin [12]. Despite all this, there are cases in which administration in high quantities was correlated with states of anxiety, palpitations, facial redness, and headaches [25]. (6) Glucuronolactone is used to improve joint functionality and lower cholesterol and triglycerides [26]. It is also a precursor for ascorbic acid [27].

Therefore, the present study aims to review the relationship between the selected group of B vitamins, glucuronolactone, and the immune system in terms of bioavailability, doses, and efficiency. To this end, multiple selection criteria were applied to the searched databases including: (1) Studies on the B vitamin bioavailability; (2) Research on the long-term impacts of energy beverages and complex B vitamins in humans; (3) Studies on the short-term effects of energy drinks as well as different B vitamins in animals; (4) Manuscripts showing the advantages of consuming water-soluble vitamins; (5) Case studies or cohorts demonstrating the risks of consuming vitamin overdoses in both humans and animals; (6) The relationship between specific B vitamins and risk of cancer; and (7) Glucuronolactone absorption and its possible effects on human health.

Relevant terms and keywords, such as glucuronolactone, water-soluble vitamins and their physiologic function, vitamin overdoses and toxicity. All these terms were used to adhere to the review criteria. Additionally, the review covered a period from 1998 to 2023.

## 2. The Absorption of Selected B Vitamins (B2, B3, B5, B6, B12) and Glucuronolactone

Vitamin B2 (VB2) or riboflavin (RF)—In recent years the mechanism and regulation of the intestinal absorption of water-soluble vitamins has been more extensively addressed. It is well established nowadays that this process takes place via certain carrier-mediated pathways. Transcriptional and/or post-transcriptional mechanisms are involved in the control of these activities, which are influenced by multiple conditions and factors [28]. Previous research has demonstrated that the RF uptake process in the small intestine and in the colon is specific and carrier mediated. It has also been demonstrated that the human gut expresses all three of the known RF transporters (RFVT-1, 2, and 3; products of the SLC52A1, SLC52A2, and SLC52A3 genes, respectively), with RFVT-3 expression being the most prevalent [29,30]. Two sources of RF are available to the human gut: the first is from a dietary source and is absorbed in the small intestine; the second is from a bacterial source and is supplied by the normal microflora of the large intestine and absorbed in that area of the intestinal tract [31]. There is a significant amount of free (absorbable) form RF generated by bacteria. This source may be beneficial to the body’s overall vitamin intake, particularly to the local colonocytes’ health and cellular nutrition. There is not much free RF in the diet in the form of an isoalloxazine ring attached to a ribitol side chain. Flavin mononucleotide (FMN) and flavin adenine dinucleotide (FAD) are two major coenzyme forms of dietary RF [32] (Figure 1).

Vitamin B3 (VB3) or niacin—Following ingestion, the gut enzymatically transforms nicotinamide adenine dinucleotide (NAD) and nicotinamide adenine dinucleotide phosphate (NADP) into nicotinamide, which is then absorbed together with nicotinic acid (NA) [33]. Nicotinamide is absorbed fast, and at low concentrations, it is promoted by Na^+^-dependent diffusion. Even at extremely high dosages of 3–5 g, niacin is almost entirely absorbed through passive diffusion [34]. Through a different process, intestinal bacteria produce VB3 mostly from the amino acid tryptophan, just like other higher species [35]. Following thorough bacterial genetic evaluation, there are only 4 members of the *Bacilli class*, 44 members of the *Clostridia class* and 29 members of the *Proteobacteria class* that have the capacity to synthesize niacin [36]. Due to the fact that *Ruminococcus lactaris*, *Prevotella copri*, and *Bacteroides fragilis* demonstrate similar capabilities of VB3 synthesis, it has been demonstrated that VB3 can also be synthesized in the stomach [37]. The intestinal epithelial cells of mice and humans have a controlled, selective, and effective system for absorbing VB3 (Figure 1). All bodily tissues receive niacin, which is then transformed into the coenzyme NAD, which is its primary active form. While nicotinic acid and nicotinamide both enter erythrocytes by assisted transport to create a circulating reserve pool and promote these cells’ function, niacin in both forms enters cells by simple diffusion [14]. VB3, produced by bacteria, helps in nourishing colonocytes locally and preserves the shape of intestinal stem cells [38]. When VB3 levels are insufficient, inflammatory bowel disorders including ulcerative colitis relapse. Additionally, the role VB3 in diminishing inflammation was also observed. The mechanism involved is mediated by stimulation of the PGD2/DP1 signal in endothelial cells, which inhibits vascular permeability in intestinal tissues and thus reduces inflammation [39].

Vitamin B5 (VB5) or pantothenic acid—Two sources of VB5 are available to the digestive tract: from dietary and from bacterial sources [31] (Figure 1). Through active transport (and potentially simple diffusion at higher dosages), pantothenic acid is absorbed in the colon and immediately enters the bloodstream [12,33,40]. Nevertheless, intestinal cells absorb panthetheine, the dephosphorylated form of phosphopantetheine, and transform it into pantothenic acid before delivering it to the circulation [2]. Though its contribution to the total amount of pantothenic acid absorbed by the body is unknown, the intestinal flora also creates pantothenic acid [41]. The sodium-dependent multivitamin transporter (SMVT, SLC5A6) is an active transporter of free pantothenic acid at low luminal concentrations [42]. Red blood cells transport pantothenic acid throughout the whole body [41]. However, smaller amounts are present as free pantothenic acid or as an acyl carrier protein. Since VB5 is the key precursor for the biosynthesis of coenzyme A (CoA), the majority of pantothenic acid in tissues is found as CoA [43].

Vitamin B6 (VB6) or pyridoxine—Dietary VB6 is absorbed by the small intestine after being transformed from pyridoxal 5′-phosphate (PLP) or pyridoxamine 5′-phosphate (PMP) to free VB6 by endogenous enzymes such as pyridoxal phosphatase. While VB6 can be absorbed through carrier-mediated and acidic pH-dependent transport, no mammalian species have been found to express an intestinal pyridoxine transporter [44]. Free VB6 is absorbed, enters the bloodstream, and it is subsequently transformed back into PMP or PLP. Bacteria use either the salvage or the de novo mechanisms to produce VB6 in the mammalian gut (Figure 1). These biosynthetic processes enable microorganisms to synthesize VB6, including *Bacteroides fragilis* and *Prevotella copri* (Bacteroidetes), *Bifidobacterium longum* and *Collinsella aerofaciens* (Actinobacteria), and *Helicobacter pylori* (Proteobacteria) [45]. While the large intestine still absorbs some dietary and bacterially generated VB6, the majority of vitamins received from food are absorbed in the small intestine [46]. The intestinal absorption of VB6 occurs from two sources: food consumption in the small intestine and absorption of vitamin B6-producing bacteria in the large intestine [47]. After absorption, multicellular organisms transfer VB6 to various host organs, whereas auxotrophic prokaryotes and single-cell eukaryotes import it from their environment [46].

Vitamin B12 (VB12) or cobalamin—Bacteria, yeasts, and probably some types of algae are the only microorganisms that are able to produce VB12. Several bacteria able to synthesize vitamin B12 are found to inhabit the intestines of monogastric animals and ruminant herbivores [48]. These processes include the involvement of three proteins: transcobalamin (TC), haptocorrin (HC), and intrinsic factor (IF), together with the corresponding membrane receptors [49]. The pancreas, small intestine, and stomach are all connected to these processes. Also, the bacteria inhabiting the colon have the ability to produce VB12, but they do not contribute significantly to the total amount of VB12 [50]. Intrinsic factor (IF) is a secreted glycoprotein produced by parietal cells in the human stomach mucosa and is in charge of the absorption of the vital vitamin cobalamin (Cbl) by the ileum. Gastric intrinsic factor (GIF) is the gene that encodes for IF. It belongs to the cobalamin transport family. It is located on human chromosome 11 (11q13) and mouse chromosome 19 [51].

Glucuronolactone—Mammals, including humans, naturally generate modest amounts of the non-toxic chemical D-glucaric acid. The D-glucuronic acid path in mammals produces the end products D-glucaric acid and D-glucaro-l,4-lactone. The oxidation of D-glucuronic acid’s lactone to D-glucaro-l,4;6,3-dilactone has been attributed to the enzyme D-glucuronolactone dehydrogenase. This dilactone is a strong beta-glucuronidase inhibitor that hydrolyzes spontaneously in aqueous solution to D-glucaro-l,4-lactone [52]. B-glucuronidase is able to hydrolyze glucuronide conjugates and it is found in the bloodstream of most likely all vertebrate organs. Colonic microflora also generates this enzyme. Higher B-glucuronidase activity has been linked to a higher risk of developing several types of cancer, especially those that are hormone-dependent, like prostate and breast cancer. It has been discovered that the stomach forms D-glucaro-l,4-lactone from provided D-glucarate salt. It is then absorbed from the intestinal tract, passes through the blood to various internal organs, and is eliminated either by bile or by urine [53].

## 3. The Riboflavin/Vitamin B2 (RF/VB2) Dose, Efficiency and Bioavailability

Sources and bioavailability: From a chemical perspective, riboflavin (RF) is 7, 8-dimethyl-10-ribityl-isoalloxazine, made up of a glucose side chain called ribitol and a flavin isoalloxazine ring [54]. Another name for RF is a water-soluble, heat-stable important VB2. While cooking does not affect its radiofrequency levels, light exposure can. Numerous foods and natural sources, including milk, organ meats (mostly found in calf liver), eggs, fish, nuts, some fruits and legumes, wild rice, mushrooms, dark green leafy vegetables, beer, cheese, and nutritional supplements, are known to contain RF [55]. It has been observed that RF is characterized by a relatively high bioavailability capability from the diet, at around 95% [12]. RF is extremely important to preserve human good health. Thus, in order to prevent ariboflavinosis, which results in cheilitis, a sore tongue, and a scaly rash on the scrotum or vulva, oral RF administration via a nutritious diet is necessary.

Within the cells, RF undergoes phosphorylation to produce flavin mononucleotide (FMN), which is then further broken down into flavin adenine dinucleotide (FAD) [56]. As cofactors in energy metabolism, FMN and FAD are essential for co-enzyme function in a variety of oxidation and reduction reactions in all aerobic forms of life. The products of more than 90 genes, or hundreds of distinct flavoenzymes, collectively constitute the human flavoproteome. Of these, 84% use FAD as a cofactor, whereas only 16% rely on FMN. Some less common instances are methionine synthase reductase, which requires both cofactors concurrently [57].

Regarding the immunological effects, supplemental diets and treatments for inflammatory diseases such as angular cheilitis, glossitis, sepsis, cataracts, and migraine headaches all include recommendations of RF supplementation [58,59]. The mechanisms involves suppressing inflammatory cytokines such as NO, IL-1, IL-1β, IL-6, and IFN-γ [32] (Table 1). Likewise, RF contributes to decreased oxidative stress by upregulating the production of catalase and inducible nitric oxide synthase (iNOS) [60]. RF is necessary for the optimization of reactive oxygen species (ROS) in the fight against bacterial infections caused by *Staphylococcus aureus* and *Listeria monocytogenes* [61] (Table 2).

From a metabolic perspective, the main biochemical pathways affected by VB2 involve lipid metabolism. Oxidoreductases, monooxygenases, dehydrogenases, transferases, hydroxylases, and oxidases are flavin coenzymes dependent. These facilitate biochemical pathways that desaturate essential fatty acids, produce phospholipids, and synthesize cholesterol, sphingosine, and steroid hormones. Moreover, apoptosis-inducing factor (AIF), is dependent on FAD and induces caspase-independent programmed cell death in the intermembrane space of mitochondria [98]. In the direction of FAD transport, an even more thoroughly studied FAD-dependent mechanism regulates protein folding in the endoplasmic reticulum (ER) [99]. FAD-dependent oxidases regulate the expression of genes involved in energy metabolism in the nucleus mediated by participating in chromatin remodeling and epigenetic processes [100]. Furthermore, RF exerts a key function in the metabolism of pyridoxine, the recycling of folate and VB12, and therefore in one-carbon metabolisms. It also plays a crucial role in the biosynthesis and control of coenzyme A, coenzyme Q10, heme, steroids, and thyroxine [32,101,102]. Skin dyscrasias resembling those seen during essential fatty acid deficiency are expressed by people with mild RF insufficiency. Thus, hepatic mitochondrial fatty acid oxidation is significantly impaired [103]. Regarding the electron transport chain, RF is essential for regulating the 2- and 1-electron acceptor/donor complexes. Four complexes (I–IV) which are formed by various carrier molecules are embedded in the inner membrane of the mitochondria. Flavoprotein reductases (dehydrogenases) and electron-transferring flavoproteins are found in complexes I and II, while ubiquinone (coenzyme Q10) and different cytochromes are found in complexes III and IV. A strong correlation between complex I anomalies and mitochondrial illnesses was observed. This link primarily appears in neurodegenerative diseases [104]. The normal functioning of lipid metabolism, energy metabolism, redox balance, and the metabolism of pharmaceuticals and xenobiotics are all directly impacted by the nutritional status of RF [18,104].

Obviously, the way that food is processed can affect RF’s bioavailability. Therefore, the vitamin may be eliminated by blanching, grinding, fermenting, and extruding. Because riboflavin photo-oxidizes when exposed to UV light, a significant amount of it is lost when fruits and vegetables are sun-dried. Oral riboflavin obtained by diet or most multivitamin supplements seldom shows toxicity or produces negative outcomes. RF is absorbed mostly in the small intestine and additionally, few quantities are absorbed in the large intestine [105]. It uses certain transport-mediated pathways that are facilitated by three solute carrier family 52A members. They belong to the RF transporters, RFVT1, RFVT2, and RFVT3, and each of them has unique kinetical and functional characteristics [30]. Failure to convert RF into FMN or FAD can result in free RF, which the kidneys release and secrete into extracellular fluids through the ABCG2 transporter. This can also cause urine to become yellow. In addition, tryptophan-riboflavin adduct, which forms from the metabolic path of an excess of RF, has been demonstrated to exert cytotoxic and hepatotoxic effects. It is especially harmful to lens proteins and the retina, which are constantly exposed to light [106].

On the one hand, using carcinogenic inducers, RF reduced the duration of liver cancer in animal studies. Reduced levels of antiapoptotic proteins and enhanced expression of apoptotic genes were the mechanisms suggested to explain this effect [107]. Research using N-nitrosomethylbenzene to induce the development of esophageal cancers in RF-deficient rats revealed raised genomic instability linked to chronic inflammation, as compared to control animals fed a standard diet containing RF [108]. In animal experiments, RF prevented lung metastases from melanoma [62]. On the other hand, it has been reported that RF functions as a coenzyme for several cytochrome P450 enzymes, serving to prevent DNA damage caused by several kinds of carcinogens [109]. Surprisingly, in both animal models and investigations in cell systems, RF deprivation has been linked to the prevention of tumor growth [110]. Moreover, research findings indicate that the proliferation, invasion, and migration of cancer cells were significantly enhanced by high-dose RF supplementation [63]. Targeting enzymes that include flavonoids to inhibit mitochondrial respiration also eradicated cancer stem cells (CSCs) [111]. Elevated serum RF concentrations were linked to a significantly higher risk of colorectal cancer (CRC), according to Ma et al. [64]. After controlling a number of variables, including sex, age, history of polyps, medical conditions, medications, Body Mass Index (BMI), and another CRC-related nutritional status, the relationship between RF and CRC risk persisted and showed a dose–response relationship [64].

Further investigations on RF are necessary to identify potential protective or side effects in new cancer types and situations, as the mechanisms by which it may modulate cancer risk still remain unclear.

## 4. The Effects of Niacin/VB3 Bioavailability, Dose and Efficiency

Sources and bioavailability: All B vitamins are water-soluble. Many different types of food contain niacin. A number of foods derived from animals, such as fish, chicken, and beef, contain 5–10 mg of niacin per serving, mostly in the highly accessible forms of NAD and NADP [40]. Plant-based foods, primarily in the form of nicotinic acid, offer 2–5 mg of niacin per serving, with nuts, legumes, and grains contributing the most. However, naturally occurring niacin is mostly associated with polysaccharides and glycopeptides in various grain products, which reduces its bioavailability to just 30% [112]. In the US and many other nations, niacin is added to a wide variety of breads, cereals, and baby formulae. Foods that have been enriched and fortified with niacin have been added in its free form, making it extremely bioavailable [12]. Since tryptophan can be converted to NAD, mostly in the liver, when it is present in amounts greater than those needed for protein synthesis, it becomes another food source of niacin [113]. The ratio of 1:60, or 1 mg of niacin [NAD] from 60 mg of tryptophan, is the most widely used estimate of the efficiency of tryptophan conversion to NAD [40].

Firstly, niacin is metabolized in the hepatic tissue, and thereafter it is delivered to all body tissues. Niacin, serves as a component in the formation of nicotinamide-adenine-dinucleotide (NAD), which is crucial for the redox processes that take place in living cells [114]. VB3’s main use is in medicine, specifically in the treatment of pellagra and alcoholism [115], which results in a yearly manufacturing capacity of more than 22,000 tons globally [114]. The majority of VB3 is currently generated chemically, with a high energy need and the creation of hazardous effluents, by the ammoxidation and oxidation of pyridines in high-pressure and alkaline circumstances [116]. In the past, Pellagra was caused by niacin insufficiency due to malnutrition, which is far less common today. Niacin deficiency has also been linked to elevated fasting glucose levels in diabetics and an increased risk of developing diabetes in non-diabetic persons [117,118,119]. Niacin, though, can result in more severe side effects when taken in higher dosages. These include multiple organ failure, hypotension, and hepatotoxicity, among others [120].

Niacin is the coenzyme for many metabolic reactions. In the body, it undergoes transformation reactions through which it reaches its active forms nicotinamide adenine dinucleotide and nicotinamide adenine dinucleotide phosphate [121]. It is involved in lipid and carbohydrate metabolism, and in redox intracellular reactions. In this way, it has a strong action against ROS [65]. Among all forms of niacin, the only one that can act as a reactive component is nicotinamide [122]. The first pharmacological component used for decreasing total cholesterol concentration was niacin. Moreover, it is capable of increasing HDL-C concentration and decreasing LDL-C and triglycerides (TGs) concentrations [66]. It has been observed that niacin has a protective role in cognitive decline specific to Alzheimer’s disease [67] (Table 2).

Regarding sports performance, evidence accrued lately is contradictory, since several recent studies show no effect on sportive performance parameters after niacin administration in human subjects as compared to previous studies [123]. On the contrary, in murine models, a diet rich in nicotinic acid (NA) combined with endurance exercises (5 times a week) over a period of 42 days changed the composition of the muscle fibers, thus enhancing endurance performance [124]. Decreasing plasma TG concentration may be the result of NA receptor-mediated inhibition of lipolysis in adipose tissue. In this way, the availability of free fatty acids (FFAs) as a substrate for the synthesis of TGs decreases [125]. Interestingly, the explanation is not complete, because it has been observed that during chronic treatment with niacin, the TGs plasma levels are higher than before [126]. Considering this, the modulating effects of niacin on the availability of FFAs were also studied in skeletal muscles that do not express NA receptor, hydroxycarboxylic acid receptor 2. These muscles, due to their large mass and thus preferential energy consumption, in resting conditions of fatty acids, are able to decrease the concentration of plasma TGs. All these statements confirm the results of the study in obese rats that treatment with normal doses of niacin can change the phenotype of skeletal fibers, and thereby affect muscle metabolism [127]. Skeletal muscle has great plasticity in terms of phenotype in response to internal as well as external stimuli. Unfortunately, in obese individuals, excess fat decreases the percentage of type I, oxidative fibers which contain more mitochondria. At the same time, the number of glycolytic fibers and type II fewer oxidative ones increases. An inverse effect is associated with weight loss [128,129]. Interestingly, it was observed that the administration of niacin in moderate doses in obese rats was able to counteract the phenotypic changes induced by the excess of adipose tissue. Due to the fact that type I fibers are richer in mitochondria, the metabolism of FFAs is more efficient and thus results in a much larger amount of ATP. This metabolic phenotype induced by niacin treatment is also confirmed by stimulation of the expression of genes involved in FFAs metabolism and in oxidative phosphorylation at this level [127]. Therefore, in this way, the increased use of FFAs by the muscles through the growth of type I fibers explains why the treatment with niacin decreases the concentration of plasma triglycerides [124]. The same effects are produced by regular endurance exercises that are able to transform glycolytic type II fibers into type I fibers [130].

If the recommended doses of niacin have beneficial effects on the body, what about overdoses? Among the most known effects are hepatotoxicity, hypotension, and damage in different organs [120]. Considering that energy drinks contain high doses of niacin, isolated reports of acute hepatitis start to emerge [68]. The most well-known side effects are linked to vasodilation changes caused by G-protein-coupled receptor 109A sensitivity to niacin in the Langerhans cells of the epidermis [131]. As a consequence, some prostaglandins are activated. Hereafter, flushing appears in the chest, arms, and in the face. It starts to diminish after 60 min [132]. The hepatotoxicity of niacin varies from a significant increase of transaminases to acute liver damage. The increase of transaminases can indicate the destabilization of the hepatocyte membrane [133]. Ingestion of approximately 3 g per day is correlated with hepatotoxicity. However, the ingestion of 30 mg per day is sufficient to produce face flushing [69]. Heemskerk’s study shows that chronic administration of niacin for 15 weeks in mice produced insulin resistance. By regulating the adipocytes gene encoding the cAMP-degrading enzyme phosphodiesterase 3B (PDE3B), niacin induced an increase in the β-adrenergic response. In this way, part of the insulin resistance mechanism produced by long-term niacin administration can be explained, which is still unknown [70] (Table 2). In Zhang’s study performed on obese mice (via diet), long-term administration of niacin for 6 weeks determined the morphological change of the islets of Langerhans as a result of the decrease and impairment in β pancreatic cell mass. Normally, these changes were biochemically confirmed by the increase in blood glucose concentration, decrease in insulin secretion, and glucose tolerance. This impairment of β cell mass may be caused by the fact that niacin can enhance this cell’s lipotoxicity, partially through an upregulation mechanism of GPR109A (G protein-coupled receptor) or Niacr1 (Niacin receptor 1) and PPARγ2 (Peroxisome proliferator-activated receptor γ2) [134]. Dietary administered VB3 may exert an impact on DNA repair, genomic stability, and the immune system, ultimately affecting cancer risk and chemotherapy side effects in the cancer patient. The NAD VB3 form takes part in several different ADP-ribosylation processes. At least seven distinct enzymes combine to form the negatively charged polymer known as poly(ADP-ribose), which is primarily found on nuclear proteins. The majority of polymer synthesis is carried out by poly (ADP-ribose) polymerase-1 (PARP-1), which also plays crucial roles in DNA damage responses, including repair, preservation of genomic stability, and signaling activities for stress responses like apoptosis. Additionally, mono (ADP-ribose), which is frequently found on G proteins and plays unclear roles in signal transduction, is made using NAD [135] (Figure 2).

The relationship between niacin (VB3, nicotinic acid) and cancer risk is mostly unknown [136,137]. One of the most well-known signs of severe niacin deficiency in humans is increased skin sensitivity to sunlight [138]. This condition is linked to deficits in the ability to respond to ultraviolet (UV) damage and low levels of NAD [139]. Niacin and its derivative, nicotinamide (niacinamide), when used topically or orally have been shown to decrease UV-induced immunosuppression, which has been proposed as a potential risk factor for skin cancer in a variety of studies including animals and humans [140,141,142]. The mechanism linking niacin with the risk of skin cancer, particularly melanoma and keratinocyte carcinoma is still not well understood. Interestingly, sex hormone synthesis may be enhanced by niacin, which could promote the growth of melanocytes [143]. Because melanocytes are known to have androgen and estrogen receptors, earlier research showed that cutaneous nevi could be a sign of the amount of plasma estrogen hormone [144]. The risk of cutaneous melanoma may rise with increased estrogen exposure [145]. Future research is required to determine the underlying mechanisms behind the relationship between niacin intake and sex hormone levels, as there is currently little knowledge in this area.

## 5. Pantothenic Acid/VB5 Bioavailability, Dose and Efficiency

Natural sources of VB5 include a variety of plants and animals (such as beef, poultry, eggs, milk, vegetables, and whole grains). CoA and phosphor-pantetheine make up around 85% of the pantothenic acid found in food [33,40]. Additionally, it is synthetically added to food. The majority of bacteria, including *Escherichia coli, Salmonella typhimurium*, and *Corynebacterium glutamicum*, produce pantothenate from aspartate, an amino acid and a step in the biosynthesis of valine [146]. VB5 insufficiency is relatively uncommon because so many foods contain this vitamin. But it can also appear in people who are very malnourished. Finding the consequences that are unique to VB5 insufficiency can be difficult because a person with VB5 deficiency frequently also has shortages in other nutrients. In an experimental study on VB5 insufficiency, symptoms like exhaustion, headaches, malaise, personality changes, numbness, muscular cramps, paresthesia, muscle/abdominal cramps, nausea, and poor motor coordination were linked to the condition [147]. A deficiency in pantothenic acid is also likely in people with mutations in the pantothenate-kinase 2 (PANK2) gene. The PANK2 mutations decrease pantothenate-kinase 2 activity, which might diminish the conversion of pantothenic acid to coenzyme A (CoA) and result in lower CoA levels. Pantothenate kinase-associated neurodegeneration (PKAN) is also brought on by PANK2 gene mutations. The “eye of the tiger” sign, which is formed by an accumulation of iron in the brain, is a characteristic feature of people with PKAN [148]. Along with other symptoms that might differ greatly from case to case, this disease also exhibits a progressive mobility problem [149].

The synthesis of acetylcholine and melatonin requires the presence of pantothenic acid. By promoting the synthesis of acetyl-CoA, coenzyme A is also used as a mechanism for moving carbon atoms inside the cell. Consequently, pantothenic acid plays a crucial role in the conversion of pyruvate to acetyl-CoA and a-ketoglutarate to succinyl-CoA, both of which are required for the stimulation of the tricarboxylic acid cycle in the sense of energy [150,151]. Thus, it plays an important role in lipid metabolism. Also, the metabolic pathways in white blood cells actively convert VB5 into coenzyme CoA; interruption of these pathways can lead to unexpected inflammatory reactions. Among others, oxidative stress management cellular adhesion, and polynuclear effectiveness are affected by pantothenic acid metabolism [71]. Supplementing with VB5 aided in the clearance of *Mycobacterium tuberculosis* and encouraged the generation of antibacterial cytokines in both isolated macrophages and infected animals [72]. Although VB5’s direct contribution to the etiology of atherosclerosis has not yet been fully understood, it may support the inflammatory process by raising CoA levels and encouraging the synthesis of glutathione (GSH), which lowers oxidative stress [73,74].

Pantethine, a derivative of VB5, is used as a treatment in patients at low to moderate risk of cardiovascular disease decreased cholesterol fractions as well as total cholesterol [75] (Table 2). The effects of VB5 depend primarily on the dose, thus large doses can cause diarrhea or functional disorders of the digestive tract [150]. Nonetheless, it is still unknown how supplemented VB5 interferes with a wide variety of drugs. Their combined administration is characterized by hypersensitivity or allergy to the drug. Moreover, in a medical report, it was observed that the consumption of VB5 is correlated with cognitive disorders [152]. In energy drinks, VB5 dose is 37.3% higher than the daily recommended dose [19] (Figure 3).

As a substrate for CoA synthesis, VB5 is involved in the normal functioning of brain cells. This is due to its role in the synthesis of different amino acids, phospholipids, and fatty acids [74,153]. Moreover, the synthesis of different steroid hormones as well as some chemical mediators is affected by VB5 via CoA signaling pathways [74]. Interestingly, the high intake of VB5 increases the activity of CoA, which can lead to an exacerbated production of immunologically active compounds that play a role in the body’s defenses, such as acute response proteins associated with inflammation, pro-inflammatory cytokines, and adhesion molecules. This fact can have repercussions on the human body’s immune response [152]. Also, supplementation with pantothenic acid for 16 weeks did not change the composition of skeletal muscle in acetyl-CoA concentration as well as exercise performance in male cyclists [154]. There are studies that report only mild adverse reactions after the treatment of acne and rheumatoid arthritis with pantothenic acid. More recently, pantothenic acid derivatives such as pantothenol, dexpanthenol, and hopantenic acid are used as medications. Thus, in Japan, they were used in compartment anomalies and mental retardation, but without having any benefic effect [155]. The most dangerous reactions were recorded after treatment with calcium hopantenate, some even fatal [78]. These included semicoma, Reye-like syndrome, and encephalopathy [77]. Gastrointestinal disorders and liver dysfunctions have been reported in the elderly [78,79].

In a human study including 38 patients, the serum levels of VB3 in breast cancer patients were lower than those in the healthy control group and the breast benign disease groups, while the levels of VB1 and VB5 in the serum of breast cancer patients and patients with benign breast diseases were higher than those in the healthy control group [156]. In a study in which responses to PD-1-targeted immunotherapy were determined in a small sample of melanoma patients, they positively correlated with plasma levels of VB5. Additionally, mice supplemented with VB5 have more success with PD-L1-targeted cancer immunotherapy, and T cells cultured in vitro with CoA have improved antitumor activity when transplanted into mice. According to these findings, VB5 is yet another B vitamin that promotes anti-cancer immunosurveillance [76]. On the other hand, greater baseline levels of plasma VB5 are associated with an increased risk of all-cause death in Chinese patients with hypertension, particularly in older adults and those with sufficient folate levels. If the findings are validated, they could influence the development of new therapeutic and dietary recommendations as well as therapies that would maximize VB5 levels [157]. Furthermore, a 12-month prospective investigation discovered a strong correlation between higher VB5 intake and higher rates of genome damage, a biomarker for a higher risk of cancer [80] (Table 2). The function of pantothenic acid has been better understood because of research on a family of proteins expressed by vascular non-inflammatory molecule (vanin) genes. Pantetheine is produced by CoA catabolism and vanins, also referred to as pantetheinases, break down pantetheine into pantothenic acid and cysteamine, the latter is associated with exacerbated inflammation [71]. Mice lacking in vanin-1 are not susceptible to paraquot or γ-irradiation-induced apoptotic oxidative tissue damage [158]. The effects of VB5 must be assessed using its recognized metabolism, which has so far brought attention to the pro-inflammatory qualities of cysteamine [159]. Considering the significance of CoA, more research seems necessary to ascertain how they affect inflammatory processes and cancer.

## 6. The Effects of Pyridoxine/VB6 Bioavailability, Dose and Efficiency

Sources and bioavailability: The body needs the micronutrient VB6 for optimal functioning. Many different foods contain VB6 [40]. The richest food sources containing vitamin B6 include fish, potatoes and other starchy vegetables, cow liver and other organ meats, and fruit (other than citrus). The majority of individuals’ daily intake of vitamin B6 in the US comes from starchy vegetables, meat, poultry, fortified cereals, and some non-citrus fruits [160]. In a varied diet, approximately 75% of vitamin B6 is easily reached [12].

The term “vitamin B6” refers to a number of chemically related substances, the most prevalent of which being pyridoxine, which is also the substance present in commercially available vitamin supplements. The generation of neurotransmitters, amino acid metabolism, glucose metabolism, lipid metabolism, hemoglobin synthesis and function, and gene expression are just a few of the body’s enzymatic processes that require VB6 [161]. In addition, it plays a role in the biosynthesis of fatty acids, the catabolization of different compounds that are used as storage in both animals and plants, as well as the biosynthesis of plant hormones, and organelle-specific substances like chlorophyll [162]. VB6 has the ability to block the ROS release [81]. Moreover, VB6 is used for photosynthesis because it aids in the scavenging of ROS and the production of chlorophyll. It is also being suggested as a potential remedy for biotic and abiotic stress [81] (Table 2).

More than 100 coenzyme reactions involving VB6 are thought to exert an impact on colorectal cancer risk. For example, VB6’s function in one-carbon metabolism-related DNA synthesis and methylation as well as its ability to lower oxidative stress, inflammation, and cell growth are all possible ways in which vitamin B6 may reduce the risk of colorectal cancer [82]. Through a salvage mechanism that includes pyridoxal kinase (PDXK), pyridoxine 5′-phosphate oxidase (PNPO), and multiple phosphatases, the various B6 vitamers are interconverted. While the FMN-dependent PNPO oxidizes pyridoxine 5′-phosphate (PNP) and pyridoxamine 5′-phosphate (PMP) to produce piridoxal 5′-phosphate (PLP), the ATP-dependent PDXK phosphorylates the 5′ alcohol group of PN, PL, and PM to make PNP, PLP, and PMP [163]. By acting as a cofactor in the glutathione antioxidant defense system, VB6 may, on the other hand, indirectly serve as an antioxidant. PLP functions as a coenzyme in the process that converts homocysteine to cysteine through sulfuration. This pathway contributes significantly to the production of reduced glutathione (GSH) by producing cysteine. In the fight against reactive free radicals and other oxidizing agents, the GSH-dependent antioxidant system, which is composed of glutathione peroxidase (GPx), glutathione reductase (GR), and glutathione S-transferase (GST), is crucial [164,165]. The equilibrium between pro- and anti-oxidants may be upset by a decline in antioxidant enzyme activity, increasing oxidative stress and cellular damage. Studies from both epidemiological and experimental research showed a definite inverse relationship between VB6 levels and diabetes as well as a definite preventive impact of VB6 on diabetic complications. It is interesting to note that by examining the mechanisms underlying the association between this vitamin and diabetes, VB6 can be viewed as both a cause and a consequence of the disease [166].

Intake of VB6 from food is typically 1.9 mg/day in the US [83]. Also, VB6 is used for the treatment of different diseases such as type 2 diabetes, nephropathy, hypertension, and heart disease. High VB6 concentrations (>2000 mg day) have been shown to have a significant negative impact on the dorsal root ganglia [84]. There is little knowledge regarding the long-term effects of low doses administration. At doses of approximately 70 ng/mL, sensory symptoms were reported in 80% of cases [85]. In a dose of 17 ± 92 mg/day for 2.9 ± 1.9 years, bone pains, hyperesthesia, and numbness were seen in women. The time of administration was a significant factor in all these adverse effects, and it is an essential factor that determines the toxicity of this vitamin [167]. As VB6 is a water-soluble vitamin, large doses should not give rise to problems, but the toxicity is actually related to the fact that its metabolites interact with certain proteins, and the half-life of this vitamin is quite long (~15–20 days) [168]. VB6’s bioavailability is constrained by complexes formed with isoniazid, cycloserine, penicillamine, and L-dopa. A subset of women who use oral contraceptives had reduced levels of pyridoxine phosphate [169]. Phenobarbital and phenytoin blood levels may drop in response to high pyridoxine doses [161]. Some authors assume that the accumulation of VB6 takes place in the liver [170]. There is a direct relationship between B6 overdoses and nerve damage, but the mechanism remains to be elucidated. PLP metabolites seem to be responsible for this effect [86,87]. For this reason, it is essential to know the recommended daily doses, especially in specific categories of people such as children, teenagers, and pregnant women. For women, they are 1.4 mg/day, and for men, they are 1.6 mg/day. In the case of the previously mentioned categories, they are easily changed [171]. It is considered equally important to know that if the daily ingested dose of 25 mg/day is exceeded, negative effects on the nervous system begin to appear [172]. Elevated plasma levels of VB6 have been observed to be associated with peripheral and sensory neuropathy. In some cases, they have been correlated with hand numbness, absent tendon reflexes, mild motor neuropathy, and photosensitivity [84,173]. Contrary to major belief, vitamin B6 toxicity symptoms are sometimes mistaken for those of VB6 insufficiency [174]. The most probable cause of patients consuming pyridoxine levels above the safe suggested upper limit is the fact that these supplements are offered over the counter. Patients frequently are not aware of the risk caused by large doses of pyridoxine. In a published case report, a patient with peripheral neuropathy was admitted to a hospital and found that he consumed vitamin B-complex supplements at a rate of fifty times the daily recommended dosage for ten years [175]. In order to further investigate the neurotoxic consequences, one study looked at the function of pyridoxine toxicity on human cells. They discovered that pyridoxine blocked pyridoxal-5-phosphate-dependent enzymes and caused cell death in a concentration-dependent manner [172]. The symptoms of VB6 toxicity resemble those of VB6 insufficiency, suggesting that the inactive form of B6, pyridoxine, competitively inhibits the active VB6 form, pyridoxal-5′-phosphate [176]. The commonly used indicator of VB6 status, circulating pyridoxal-5′-phosphate (PLP), has been associated in epidemiological studies with an increased risk of lung cancer among other cancers [88] (Table 2). The fact that circulating concentrations of PLP are influenced by a number of factors, such as dietary or supplement intake, inflammation, serum albumin, and alkaline phosphatase levels, may account for the wide variation in estimated associations of PLP with lung cancer risk observed in studies [177,178].

The PAr index has been proposed, which is defined as the ratio 4-pyridoxic acid (PA)/(pyridoxal + PLP), in light of PLP’s limitations as a biomarker [179,180]. There is evidence that a number of inflammatory mechanisms, including PLP-catabolizing enzymes, oxidative stress, and kidney damage, might skew plasma B6 vitamin concentrations toward greater PA than pyridoxal+ PLP, which raises PAr [181]. Therefore, during inflammation and associated cellular immunological activation, PAr acts as a sign of enhanced vitamin B6 catabolism [182]. The results of two studies, the European Prospective Investigation into Cancer and Nutrition (EPIC) and the Hordaland Health Study (HUSK), which were published, indicate a possible link between PAr and the risk of lung cancer [183].

Unfortunately, the concentration of VB6 in most energy drinks is significantly high. Each type of energizer contains different concentrations of VB6 [184]. Red Bull contains 5 mg, whereas Prix Garantie Energy drink contains 1.99 mg. The consumption of energy drinks was the primary cause of extremely raised plasma vitamin B6 levels, whereas elevated plasma VB6 levels up to four times above the upper normal limit are typical in postbariatric patients and are related to regular multivitamin treatment [24].

## 7. The Cobalamin/VB12 Bioavailability, Dose and Efficiency

Sources and bioavailability: Certain bacteria and archaea can produce VB12, but neither plants nor animals can [185]. Thus, VB12 molecules found in food come from bacteria that are able to synthesize VB12, including archaea. There are two different biosynthetic pathways to obtain VB12 compounds: aerobic and anaerobic [186,187]. VB12 is considered essential for humans because it cannot be synthesized by the body [188]. Foods of animal origin, such as fish, meat, poultry, eggs, and dairy products, naturally contain VB12 [16]. Additionally, commonly available forms of VB12 with high bioavailability include enriched nutritional yeasts and morning cereals [189,190]. When the capacity of intrinsic factor is exceeded (at 1–2 mcg of VB12), the absorption is dramatically reduced, which explains why the estimated bioavailability of VB12 from food varies by VB12 dose [191]. Additionally, the kind of dietary supply affects bioavailability. For instance, VB12 from dietary supplements is about 50% more bioavailable than from food sources, and VB12 appears to be almost three times more bioavailable in dairy products than in meat, fish, and poultry [192,193,194].

Cobalamins (VB12) belong to the corrinoids family of cobalt-containing substances that are present in nature. Cobamides and cobinamides, which are cobalamin analogs that have undergone chemical modification, are also members of the corrinoid family in addition to cobalamins [195]. Furthermore, it has been shown that VB12 has two pathways by which it is absorbed into the body. One by which it is dependent on the castle factor which provides the majority of the daily requirement of B12 and an independent way, in the ileum, responsible for the absorption of 2–5% [196,197].

Nucleic acid metabolism, red blood cell formation, and restoration of myelin synthesis depend on VB12. It has a low potential for toxicity, mainly because the data in the literature are scant which does not mean that it has no adverse effects in high doses. For this reason, VB12 is not recommended for those suffering from a wide variety of diseases (different types of cancer) and statuses (pregnant women) [198]. Considering that all the active compounds are in high concentration in energy drinks, it is possible for them to accumulate, and in the end, create adverse effects [133,198]. Different methylation reactions require VB12 as a cofactor. In this way, it is involved in the conversion of methyl malonyl-CoA to succinyl-CoA as well as in the conversion of homocysteine to methionine. The two conversion pathways are important for cell division and proliferation [199]. It is involved in the prevention of cardiovascular diseases. Although these diseases are significantly decreased in children, it appears that the number of hypertensive preschool children has increased [200,201]. A diet rich in vitamin B12, in children up to 6 years can cause a decrease in blood pressure [89]. VB12 deficiency has been observed to be associated with high blood pressure [200]. In athletes, no improvement in energy was observed following VB12 consumption. However, when VB12 is consumed together with VB1 and VB6, it has been shown to exert performance-enhancing effects in pistol shooting [202]. This effect is most likely due to the increased level of serotonin obtained following the consumption of the 3 vitamins. Moreover, VB12 is associated with the improvement of cognitive processes in the elderly population [90] (Table 2).

Due to the high VB12 concentrations present in energy drinks different acute hepatitis were occasionally correlated with the consumption of these drinks [203]. A higher dietary intake of VB12 was linked to a higher risk of developing lung cancer [91]. Persistent VB12 elevation was linked to a significant prevalence of solid cancer. For patients with unexplained and persistently increased VB12 levels, solid tumors are one of the most common diagnoses. In this sense, a second measurement should be taken subsequently to confirm an unexplained increased VB12 levels. This could aid in determining which people will benefit from solid cancer screening [92] (Table 2).

It is commonly established that abnormal DNA methylation is essential to the initiation and spread of cancer [204]. A methyl donor is needed for DNA methylation and this donor is primarily supplied by the metabolites in the one-carbon metabolic pathway [205]. In fact, a number of substances including numerous B vitamins like VB6, VB9 (folate), and VB12, which are cofactors for certain enzymes in biochemical and signaling reactions to form methyl donors, are involved in one-carbon metabolism [206] (Figure 4).

Lung cancer may arise as a result of the disruption of the one-carbon metabolic pathway, which may also change DNA methylation and generate genomic instability. These two factors may accelerate the malignant transformation of cells [207]. The risk of lung cancer and VB12 have been inconsistently linked, according to previous epidemiological research. As an example, two extensive randomized, placebo-controlled trials of vitamin B12 supplementation (i.e., (1) oral treatment with folic acid—0.8 mg/d and VB12—0.4 m/d and vitamin B6—40 mg/d; (2) oral treatment with folic acid—0.8 mg/d and VB12—0.4 mg/d; (3) oral treatment with VB6—40 mg/d alone; and (4) placebo) in the Norwegian population revealed that the combination of VB12 and folic acid increased the risk of cancer overall, mainly lung cancer. It is possible that folic acid, administered for approximately 39 months, impacted the development of malignancies that were initially discrete at the beginning of the experiment or during the study, causing a higher rate of clinical manifestation and diagnosis in the folic acid groups during the period of a prolonged follow-up. After administration of an extra median period of 38 months of posttrial follow-up, vitamin B12 during a median of 39 months was linked to increased cancer incidence and cancer-related deaths. The higher incidence of lung cancer was the main cause of these findings. Moreover, drugs with both folic acid and vitamin B12 were observed to contribute to an increased mortality rate from all causes. The increased cancer mortality as well as the statistically insignificantly higher noncancer mortality were the primary causes of the second observation [208]. The Vitamins and Lifestyle (VITAL) cohort study’s findings also indicated that males who were supplemented with high levels of VB6 and VB12 had a higher chance of inducing the development of lung cancer [209]. More recently, the Lung Cancer Cohort Consortium (LC3) study found a strong correlation between elevated blood VB12 levels and a higher risk of lung cancer [210]. Nonetheless, a number of sizable cohort studies that examined a wide range of populations—including those in Australia, China, Europe, the US, and Europe—did not discover a positive or negative correlation between dietary vitamin B12 intakes and the risk of lung cancer. Asian dietary habits differ from those of Caucasians or African Americans, who were the subjects of the majority of this research [178,211,212,213]. The majority of studies demonstrate a dose-dependent correlation between dietary VB12 intake and an increased risk of developing lung cancer. This correlation is stronger in patients with adenocarcinomas, men, and those with follow-ups of equal to less than two years. This emphasizes that high-dose or high-level VB12 supplements are not helpful in lung cancer prevention programs and may potentially be hazardous, especially to men [208].

It is unclear how increased VB12 in solid cancer cases functions [214,215]. The predictive significance of increased VB12 in solid tumors raised the possibility of a connection to either the tumor bulk or the ability to proliferate [216]. According to the first theory, tumor mediators secreted by cells increase VB12’s bioavailability, which in turn encourages cancer cells to synthesize nucleic acids. The second theory concerns the granulocytic cells involved in the anti-tumor response and their secretion of haptocorrins [92].

However, different studies have shown that the prevalence of breast cancer as a whole and the likelihood of estrogen receptor (ER), progesterone receptor (PR), and human epidermal growth factor receptor-2 (Her-2) positive subtypes are reduced by high consumption of vitamins B2, B6, B12, and folic acid [156,217].

## 8. The Glucuronolactone Bioavailability, Dose and Efficiency

Sources and bioavailability: The human body generates small quantities of this organic substance. The limited human research is consistent with the glucuronolactone pharmacokinetic findings in rats, which demonstrate bioavailability, lack of accumulation, and peak plasma levels 1 to 2 h after oral treatment [40]. Rodents have the ability to synthesize vitamin C from glucuronic acid and can also convert exogenous D-glucurono-γ-lactone into vitamin C [96]. Nevertheless, this metabolic pathway is absent in primates, including humans. D-Glucurono-γ-lactone, also known as glucuronic-3,6-lactone or GGL, is a molecule that is believed to exert a large amount of benefits to human health, including the improvement of joint health, acting as an anti-inflammatory for the skin, and reducing high concentrations of cholesterol and triglycerides in the plasma. GGL is produced by the oxidation of beetroot nitric acid to glucuronic acid [26] (Table 2). Supplementing with D-glucarates, such as glucuronolactone, may help the body’s natural defense system function better to inhibit tumor promoters and carcinogens and their consequences [93]. Unfortunately, little research has been done on humans, and there is little information available right now about this chemical. As a result, it is impossible to determine whether this molecule is useful or, alternatively, dangerous. The average daily dietary exposure to D-glucuronolactone is thought to be low (1–2 mg) [96,218]. However, the discovery of unspecified renal lesions (inflammation of the renal papilla) in rats during the identification of D-glucuronolactone raised questions about the safety of using this ingredient in energy drinks. The lowest no-observed-adverse-effect limit (NOAEL) for these nephrotoxic consequences was first set at 300 mg/kg total body weight (t.b.w.)/day, but after further histological research revealed kidney inflammation, the NOAEL was ultimately determined at 1000 mg/kg total body weight (t.b.w.)/day [96]. Individual initiatives have been proposed in various countries, such as Germany and Denmark [219], promoting the standardization of EDs with maximum caffeine levels at 32 mg/100 mL; taurine levels at 4000 mg/L; and glucuronolactone contents at 2400 mg/L. This is due to the fact that exposure to D-glucuronolactone and taurine has raised safety concerns, especially in high and chronic consumption scenarios [19]. Only when their consumption via EDs with 2400 mg of D-glucuronolactone/l is restricted to 250 mL would persons weighing 60 and 80 kg present a margin of safety (MOS) 100, however in the latter instance, this is also seen when their consumption is 333 mL in 80 kg individuals [219]. These findings contradict the EFSA statement that dietary exposures at the levels present in EDs are not harmful to a person of a 60 kg body weight, even when the chronic consumption of EDs is high (350 mL/day), which was based on the NOAEL established for the toxicological effects of D-glucuronolactone (1000 mg/kg b.w./day) [96]. Last but not least, for persons with modest body weights (about 40 kg), a MOS 100 was estimated in each of the three consumption scenarios (250, 333, and 500 mL). As a result, the health hazards associated with exposure to the D-glucuronolactone contents of EDs may be anticipated [219]. There is a tremendous potential for improvement in terms of labeling, including providing information about each ingredient, particularly those that might be harmful to health, such as taurine, and D-glucuronolactone. Packaging of much smaller volumes should be promoted since doing so would help to moderate the amount of exposure to the various active ingredients. Additionally, in order to reduce exposure to the various active ingredients, some of which are psychoactive, it is advised to comply with the industry’s commitment to only market packages containing no more than 250 mL, as stated in the Scientific Committee of the Association of Southeast Asian Nations (AESAN) report, as well as researching the possibility of ceasing the marketing of 500 mL packages [220]. According to the EFSA, the food additives used in energy drinks have been found to be harmful when ingested in excess of the permissible amount [96]. However, the precise components and dose that caused side effects were not thoroughly examined or recorded. Over the course of 21 days of treatment, young rats were used to evaluate and study the neurobehavioral toxic effects of food additives when given oral doses of glucuronolactone 5 mg/kg per os (p.o.), taurine 8 mg/kg p.o., gluconolactone 84 mg/kg p.o., and a combination of the three food additives [10]. When administered together, taurine and glucuronolactone significantly altered neurotransmitter levels more than when given separately, which suggests a higher risk of neuronal toxicity, modulation, and injury [221]. These modifications provide evidence of neurobehavioral impairments [10]. It has been observed that glucuronolactone can exert negative effects on endothelial cells and platelet functions [97] (Table 2).

Clinically, due to serious adverse effects, using glucocorticoids alone to treat hepatitis without the help of liver protection and antiviral medicine is rather risky [222]. The Merck Index states that glucuronolactone, the most widely used hepatoprotective drug in some nations like China, is utilized as a common detoxification product [94,95]. Thus, in different investigations, glucuronolactone was used accordingly. It was demonstrated that while using glucuronolactone alone had no effect on slowing the progression of hepatocarcinoma, however when used in combination with hydrocortisone it exerted a protective effect in the early stages of the disease [223]. The anticancer properties of D-glucaro-l,4-lactone and its precursors are partially attributed to modifications in steroidogenesis, which are associated with adjustments in the hormonal milieu and the proliferative state of the target organs. D-glucarates cause apoptosis in addition to inhibiting inflammation and cell division. One way to support the body’s natural defense mechanism against carcinogens and tumor promoters and their effects is to supplement with D-glucarates [93].

## 9. Conclusions and Future Perspectives

The role of vitamins is unanimously crucial in the division, proliferation and development of the body. In this sense, the place where absorption occurs and the sources of synthesis in the body must be taken into consideration. Knowing all this, we can anticipate what happens in the case of either avitaminosis and/or overdoses. Moreover, understanding the mechanisms behind them is crucial. Both conditions affect the immune system functions, particularly in terms of how pro- or anti-inflammatory compounds are released. Furthermore, because there is a strong association between dietary requirements and age, gender, and health state, these factors must be considered. Thus, even if the doses of different vitamins are small, the fact that they are taken in combination with other vitamins can affect hyperabsorption and finally hypervitaminosis. And lastly, the microbiota in the gut plays a significant role.

RF reduced the course of liver cancer in animal experiments by utilizing carcinogenic inducers. The mechanisms proposed to explain this effect were decreased amounts of antiapoptotic proteins and increased expression of apoptotic genes [107].

It is interesting to note that niacin may boost the production of sex hormones through stimulating the proliferation of melanocytes [143]. In addition, cutaneous nevi may be an indicator of the level of plasma estrogen hormone since melanocytes are known to have androgen and estrogen receptors [144].

It is relevant to consider that consuming too much VB5 diet increases CoA activity, which may raise the body’s intake of immunological compounds connected to defense. These compounds include adhesion molecules, pro-inflammatory cytokines, and acute response proteins associated with inflammation. The human body’s defenses may be affected by this fact [152].

Despite widespread misunderstandings, symptoms of VB6 toxicity may occasionally be confused with those of VB6 deficiency [174]. Without a prescription, supplements are most likely the reason why patients’ pyridoxine levels exceed than the recommended safe upper limit. Frequently patients are uninformed of the danger associated with ingesting significant amounts of pyridoxine. In one case study, a patient diagnosed with peripheral neuropathy and transferred to the hospital admitted that he had been taking vitamin B-complex supplements for ten years at a rate fifty times higher than the daily prescribed dosage [175].

Energy drink consumption has been sporadically associated to distinct cases of severe hepatitis due to the high VB12 concentration found in these drinks [203]. A higher risk of lung cancer has been associated with a larger amount of VB12 [91] consumed. At 60 months, a substantial incidence of solid tumors was associated with persistent VB12 increase. Solid tumors rate among those more often diagnosed conditions in patients presenting with chronically elevated VB12 levels that cannot be explained.

Taurine and glucuronolactone administered simultaneously modified the levels of neurotransmitters more than the two compounds administered separately, indicating a greater potential for neuronal toxicity, regulation, and damage [221].

## Figures and Tables

**Figure 1 nutrients-16-00024-f001:**
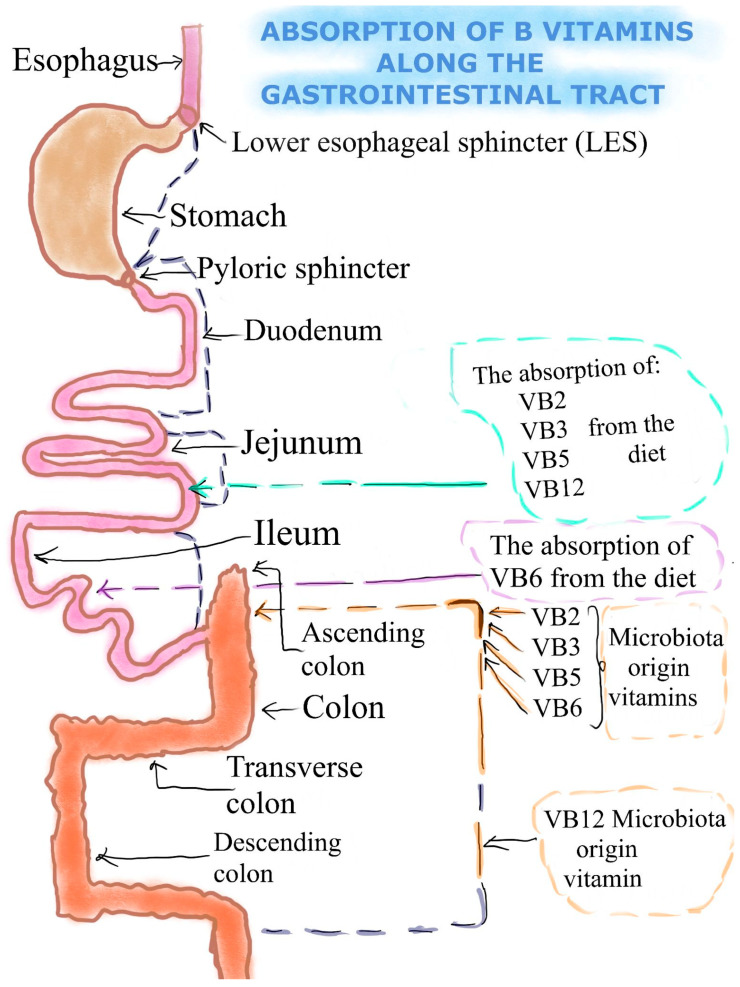
The absorption of B vitamins, VB2, vitamin B2 (Riboflavin); VB3, vitamin B3 (Niacin); VB5, vitamin B5 (Pantothenic acid), VB6, vitamin B6 (Pyridoxine); VB12, vitamin B12 (Cobalamin) along the gastrointestinal tract. There are two sources of B vitamins: diet and gut microbiota. As seen, the absorption of vitamins VB2, VB3, VB5, and VB12 from the diet takes place in the small intestine, more precisely in the jejunum. Regarding VB6, the absorption is done at the level of the ileum. In addition, in the ascending colon, there are bacteria that synthesize vitamins VB2, VB3, VB5, and VB6, while VB12 is produced by different bacteria in the descending colon.

**Figure 2 nutrients-16-00024-f002:**
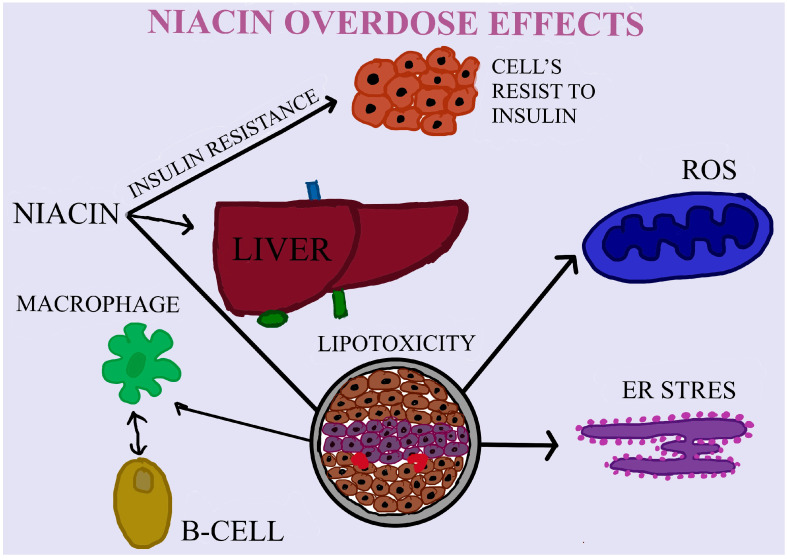
Niacin overdose effects. The most well-known side effects include harm to several organs, hypotension, and hepatotoxicity. The morphological alteration of the islets of Langerhans as a result of the impairment and decrease in β pancreatic cell mass was determined after niacin was administered for six weeks. The possible reason for this reduction in β cell mass could be because niacin increases the lipotoxicity of these cells.

**Figure 3 nutrients-16-00024-f003:**
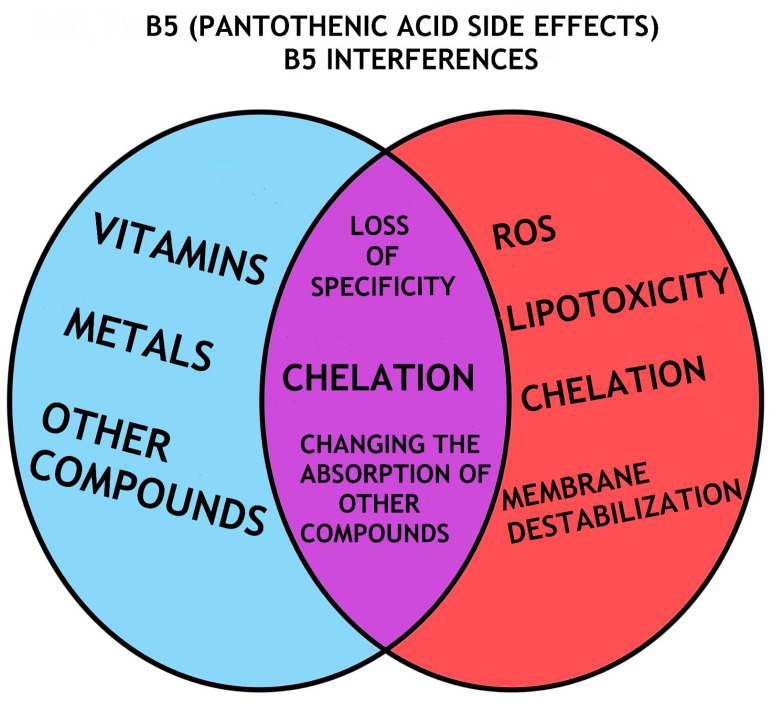
VB5 (Panthotenic acid) side effects or VB5 interferences; It is currently unclear how supplemental VB5 interacts with such an extensive variety of medicines. The combination of their administration can be defined by drug hypersensitivity or allergy. Furthermore, VB5 ingestion has been linked to cognitive impairments, according to a medical report.

**Figure 4 nutrients-16-00024-f004:**
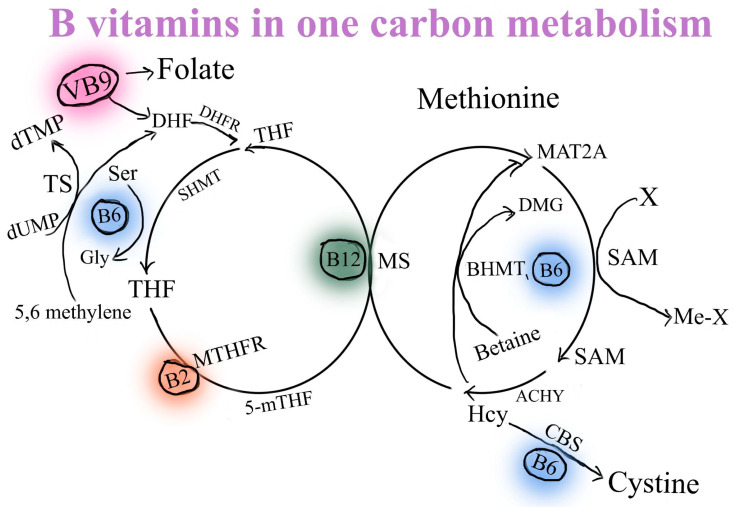
B vitamins in one carbon metabolism. The enzyme dihydrofolate reductase (DHFR) first converts dietary folate (B9) into dihydrofolate (DHF), which is subsequently reduced to tetrahydrofolate (THF). This process starts the folate cycle. Next, serine hydroxymethyltransferase (SHMT) converts THF to 5,10-methylene THF. This reaction needs B6 as a cofactor and is linked to the hydroxylation of serine (Ser) to glycine (Gly). Deoxyuridine monophosphate (dUMP) is methylated by thymidylate synthase (TS) using 5,10-methylene THF as a methyl donor, resulting in deoxythymidine monophosphate (dTMP). DHF is renewed in this phase to enable further cycling. Alternatively, methylenetetrahydrofolate reductase (MTHFR) can use B2 as a cofactor to decrease 5,10-methylene THF to 5-methytetrahydrofolate (5-mTHF). Methionine synthase (MS) is responsible for catalyzing the methionine cycle, in which 5-mTHF provides a methyl group to regenerate methionine from homocysteine (Hcy). Methionine synthase (MS) needs B12 in the form of methylcobalamin as a cofactor. Methionine adenosyltransferase 2A converts an adenosine to methionine, resulting in the methyl donor S-adenosylmethionine (SAM), which is then used by several methyltransferases (MTs) specific for methylation events involving RNA (RMT), DNA (DNMT), histones (HMT), and proteins (PRMT). S-adenosylhomocysteine (SAH), which is subsequently hydrolyzed by S-adenosylhomocysteine hydrolase (SAHH) to generate Hcy, is formed when SAM is demethylated during the methyltransferase processes. In order to produce cysteine, Hcy can also join the transsulfuration route, which is mediated by vitamin B6 and cystathionine beta synthase (CBS). Dietary betaine can function as a methyl donor in the liver. Methionine and dimethylglycine (DMG) are produced as a byproduct of the liver enzyme betaine-homocysteine S-methyltransferase (BHMT), which uses betaine from the food as a methyl donor and B6 as a cofactor.

**Table 1 nutrients-16-00024-t001:** The recommended doses for different water-soluble vitamins.

Compounds	Recommended Daily Dose for Adult Women	Recommended Daily Dose for Adult Men	Recommended Daily Dose for Pregnant Women	Recommended Daily Dose for Teenagers Girls	Recommended Daily Dose for Teenagers Boys
Vitamin B2 (Riboflavin) [12]	1.1 mg	1.3 mg	1.4 mg	1.0 mg	1.3 mg
Vitamin B3 (Niacin) [13]	14 mg	16 mg	18 mg	14 mg	16 mg
Vitamin B5 (Pantothenic acid) [14]	5 mg	5 mg	6 mg	5 mg	5 mg
Vitamin B6 (Pyridoxine) [15]	1–1.7 mg	1–1.7 mg	1.9 mg	1.2 mg	1.3 mg
Vitamin B12 (Cobalamin) [16,17]	2.4 mcg	2.4 mcg	2.6 mcg	2.4 mcg	2.4 mcg

mg, milligrams, mcg, micrograms.

**Table 2 nutrients-16-00024-t002:** The efficiency of B vitamins and the negative impact of providing high doses on the immune system.

Compounds	Efficiency in the Immune System	Negative Impacts of High Doses
Vitamin B2 (Riboflavin)	(1) antinflammatory effects in diseases such as angular cheilitis, glossitis, sepsis, cataracts, and migraine headaches all include recommendations of RF supplementation [58,59]. (2) suppresses inflammatory cytokines such as NO, IL-1, IL-1β, IL-6, and IFN-γ [32]. (3) the optimization of ROS in the fight against bacterial infections caused by *Staphylococcus aureus* and *Listeria monocytogenes* [61]. (4) in animal experiments, RF prevented lung metastases from melanoma [62].	(1) the proliferation, invasion, and migration of cancer cells were significantly enhanced by high-dose RF supplementation [63]. (2) elevated serum RF concentrations were linked to a significantly higher risk of colorectal cancer (CRC), according to Ma et al. [64].
Vitamin B3 (Niacin)	(1) it has a strong action against reactive oxygen species [65]. (2) it was the first pharmacological component used for decreasing the total cholesterol concentration. (3) it is capable of increasing HDL-C concentration and decreasing LDL-C and triglycerides (TGs) concentrations [66]. (4) it has a protective role in the cognitive decline specific to Alzheimer’s disease [67].	(1) in high doses of niacin, isolated reports of acute hepatitis start to emerge [68]. (2) ingestion of approximately 3 g per day is correlated with hepatotoxicity, while of 30 mg per day is sufficient to produce face flushing [69]. (3) chronic administration of niacin for 15 weeks in mice produced insulin resistance [70].
Vitamin B5 (Pantothenic acid)	(1) it has a crucial effect in oxidative stress management, cellular adhesion, and polynuclear effectiveness [71]. (2) it encourages the generation of antibacterial cytokines in both isolated macrophages and infected animals [72]. (3) it may support the inflammatory process by raising CoA levels and encouraging the synthesis of glutathione (GSH), which lowers oxidative stress [73,74]. (4) it is used as a treatment in patients at low to moderate risk of cardiovascular disease decreased cholesterol fractions as well as total cholesterol [75]. (5) it promotes anti-cancer immunosurveillance [76].	(1) it produced semicoma, Reye-like syndrome, and encephalopathy [77]. (2) it is associated with gastrointestinal disorders and liver dysfunctions in the elderly [78,79]. (3) higher its intake is correlated with higher rates of genome damage, a biomarker for a higher risk of cancer [80].
Vitamin B6 (Pyridoxine)	(1) it has the ability to block the ROS release [81]. (2) it may reduce the risk of colorectal cancer [82]. (3) it is used for the treatment of different diseases such as type 2 diabetes, nephropathy, hypertension, and heart disease [83].	(1) it has a negative impact on the dorsal root ganglia [84]. (2) at doses of approximately 70 ng/mL, sensory symptoms were reported in 80% of cases [85]. (3) PLP metabolites provoked nerve damage, but the mechanism remains to be elucidated [86,87]. (4) PLP has been associated in epidemiological studies with an increased risk of lung cancer among other cancers [88].
Vitamin B12 (Cobalamin)	(1) in children up to 6 years, it can cause a decrease in blood pressure [89]. (2) it is associated with the improvement of cognitive processes in the elderly population [90].	(1) a higher dietary intake of vitamin B12 was linked to a higher risk of developing lung cancer [91]. (2) persistent VB12 elevation was linked to a significant prevalence of solid cancer [92].
Glucuronolactone	(1) it improves joint health and acts as an anti-inflammatory for the skin. (2) it reduces high concentrations of cholesterol and triglycerides in the plasma [26]. (3) it may help the body’s natural defense system function better to inhibit tumor promoters and carcinogens and their consequences [93]. (4) it is used as hepatoprotective drug [94,95].	(1) it was associated with unspecified renal lesions (inflammation of the renal papilla) in rats [96]. (2) it has been observed that glucuronolactone can exert negative effects on endothelial cells and platelet functions [97].

NO, Nitric oxide; IL-1, Interleukin-1; IL-1β, Interleukin-1 beta; IL-6, Interleukin-6; IFN-γ, Interferon gamma; RF, Riboflavin; ROS, Reactive oxygen species; LDL-C, Low-density lipoprotein cholesterol; HDL-C, High-density lipoprotein cholesterol; TGs, Triglycerides; GSH, Glutathione; CoA, coenzyme A.

## Data Availability

Not applicable.

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
