# Peer review of "B Vitamins, Glucoronolactone and the Immune System: Bioavailability, Doses and Efficiency"

_nutrients, 2023, doi:10.3390/nu16010024_

Round 1

Reviewer 1 Report

Comments and Suggestions for Authors

The article „B vitamins, glucoronolactone and the immune system: bioavailability, doses and efficiency”written by Camelia Munteanu and Betty Schwartz is very comprehensive review concentrated on summarising the knowledge on the association between the B vitamins, glucuronolactone, and the functioning of immune system. The authors described the relationships considering the aspect of bioavailability, doses, and efficiency. The paper also contains the information about the most common reported effects of overdoses of the selected vitamins. The crucial mechanisms proposed to explain the effect of components of interest on the selected endpoints are clearly presented. Article is very well written and the background is properly stated.

Table 2 and graphics included in manuscript are very helpful and facilitate the reading and understanding of the paper.

Minor remarks:  I couldn’t find the title of Table 2 – please add it. The sentence „Relevant terms and keywords, such as glucuronolactone, water-soluble vitamins and their physiologic function, vitamin overdoses and toxicity.” seems to be unfinished.

Author Response

Reviewer # 1

We thank the reviewer for her/his comments. We give our response to his queries (our answers are in red; the reviewer queries are in black).

The article „B vitamins, glucoronolactone and the immune system: bioavailability, doses and efficiency” written by Camelia Munteanu and Betty Schwartz is very comprehensive review concentrated on summarizing the knowledge on the association between the B vitamins, glucuronolactone, and the functioning of immune system. The authors described the relationships considering the aspect of bioavailability, doses, and efficiency. The paper also contains the information about the most common reported effects of overdoses of the selected vitamins. The crucial mechanisms proposed to explain the effect of components of interest on the selected endpoints are clearly presented. Article is very well written and the background is properly stated.

Dear reviewer, Thank you! We are sure that with the minor remarks you suggest, our manuscript will improve.

Minor remarks:  

Table 2 and graphics included in manuscript are very helpful and facilitate the reading and understanding of the paper.

I couldn’t find the title of Table 2 – please add it.

Very good suggestion, thank you. We added the title to table 2.

The sentence „Relevant terms and keywords, such as glucuronolactone, water-soluble vitamins and their physiologic function, vitamin overdoses and toxicity.” seems to be unfinished.

Yes, you are right. We finished with a point. Thank you!

Reviewer 2 Report

Comments and Suggestions for Authors

Dear Redactors,

Thank you very much for the opprotunity to revise article „B vitamins, glucoronolactone and the immune system: bioavailability, doses and efficiency”.

The article is very interesting. It aims to review the relationship between the selected B vitamins group, glucuronolactone, and the immune system in terms of bioavailability, doses, and efficiency. The authors included studies on the B vitamin bioavailability,  research on the longterm impacts of energy beverages and complex B vitamins in humans,  studies on the shorts effects of energy drinks as well as different B vitamins in animals, manuscripts showing the advantages of consuming watersoluble vitamins, and case studies or cohorts demonstrating the risks of consuming vitamin overdoses in both humans and animals.

The article is well written. I have no specific comments.

Thanks.

Author Response

Reviewer # 2

We thank the reviewer for her/his comments. We give our response to his queries (our answers are in red; the reviewer queries are in black).

The article is well written. I have no specific comments.

Thanks.

Dear reviewer,

Thank you very much! Your answer encourages us a lot! We appreciate it!

Reviewer 3 Report

Comments and Suggestions for Authors

The manuscript is well-written and very interesting. It requires some correction of some errors.

Line 62: situation.[9]. = situation [9].

Table 1: You must center some values of vitamin B2 in the table (1.4 mg, 1.0 mg). You must fix the legend: mg, milligrams; mcg, micrograms. 

Line 116:  (RF) -In = (RF) - In

Figure 1:  You must write Figure 1 in bold.

Line 134: niacin-Following = niacin - Following

Line 166: body, [41]. = body [41].

Line 187: cobalamin -Bacteria, = cobalamin - Bacteria,

Line 199: Glucuronolactone-Mammals, = Glucuronolactone - Mammals,

Line 200: d-glucaric acid = D-glucaric acid

Line 207: b-glucuronidase = B-glucuronidase

Line 212: bioavailability,bioavailability.

Line 212: Point 1 does not have a tab indent, while point 2 does. I ask you to align the entire manuscript with the same editing format.

Table - Pag 8 of 31 - Vitamin B6 = 4) PLP has been ... You have to fix the space, in the pdf file there seems to be a double space.

Line 317: efficiencyefficiency.

Line 438: efficiencyefficiency. 

Line 528: efficiency = efficiency. 

Line 606: [163].The[163]. The 

Line 631: efficiency = efficiency. 

Line 645: [184-186] = [184-186]. 

Figure 4: You need to check the acronyms in the legend.

Line 693: [i.e., = You have to check the bracket "["

Line 693/695: 1) oral treatment with folic acid-0.8 m/d plus and VB12-0.4 m/d and vitamin B6-40 mg/d; 2) oral treatment with folic acid-0.8 m/d plus and VB12–0.4 m/d; 3) oral treatment with VB6-40 mg/d alone; = You must write more clearly

Line 722: efficiency = efficiency.

Line 743: TBW (Total Body Weight) or IBW (Ideal Body Weight) 

Line 774: p.o. = What does that mean?

Line 794: perspectives  = perspectives.

Line 1031: You must correct the bibliography (n. 89). The DOI is part of the previous article.

Author Response

Reviewer # 3

We thank the reviewer for her/his comments. We give our response to his queries (our answers are in red; the reviewer queries are in black).

The manuscript is well-written and very interesting. It requires some correction of some errors.

We are sure that after correcting these errors, our manuscript will be much better. Thank you!

Line 62: situation.[9]. = situation [9].

Thank you, we corrected it.

Table 1: You must center some values of vitamin B2 in the table (1.4 mg, 1.0 mg). You must fix the legend: mg, milligrams; mcg, micrograms.

Thank you very much for the suggestions! You're right. We modified.

Line 116:  (RF) -In = (RF) – In

You're right. We made the changes according to your observation.

Figure 1:  You must write Figure 1 in bold.

Thanks, we changed it as you suggested.

Line 134: niacin-Following = niacin - Following

Thank you, we have made the changes as you suggested.

Line 166: body, [41]. = body [41].

Thanks, we deleted the comma.

Line 187: cobalamin -Bacteria, = cobalamin - Bacteria,

Yes, you are right. We made the change.

Line 199: Glucuronolactone-Mammals, = Glucuronolactone - Mammals,

As you suggested, we made the change.

Line 200: d-glucaric acid = D-glucaric acid

We made the changes according to your suggestion. Thank you.

Line 207: b-glucuronidase = B-glucuronidase

Thank you, we changed it as you suggested.

Line 212: bioavailability, = bioavailability.

We modified it as you suggested.

Line 212: Point 1 does not have a tab indent, while point 2 does. I ask you to align the entire manuscript with the same editing format.

You are right, as you recommended we improved the aspect of our manuscript.

Table - Pag 8 of 31 - Vitamin B6 = 4) PLP has been ... You have to fix the space, in the pdf file there seems to be a double space.

We addressed and solved the space problem. Thank you.

Line 317: efficiency = efficiency.

Thanks, we put the point at the end.

Line 438: efficiency = efficiency.

Thanks, we put the point at the end.

Line 528: efficiency = efficiency.

Thanks, we put the point at the end.

Line 606: [163].The = [163]. The

Thanks, we made the changes.

Line 631: efficiency = efficiency.

Thanks, we put the point at the end.

Line 645: [184-186] = [184-186].

You're right. Now, we have finished the sentence.

Figure 4: You need to check the acronyms in the legend.

Thanks, we verified the acronyms in the legend of Figure 4. You are right, we modified where it was necessary.

Line 693: [i.e., = You have to check the bracket "["

We checked the brackets as you recommended, thank you.

Line 693/695: 1) oral treatment with folic acid-0.8 m/d plus and VB12-0.4 m/d and vitamin B6-40 mg/d; 2) oral treatment with folic acid-0.8 m/d plus and VB12–0.4 m/d; 3) oral treatment with VB6-40 mg/d alone; = You must write more clearly

As you suggested, we clarified this paragraph by adding the following text: It is possible that folic acid, administered for a median of approximately 39 months, impacted the growth development of malignancies that were initially discrete at the beginning of the experiment or during the study, causing a higher rate of clinical manifestation and diagnosis in the folic acid groups during the period of a prolonged follow-up. After administration of an extra median period of 38 months of posttrial follow-up, vitamin B12 during a median of 39 months was linked to increased cancer incidence and cancer-related deaths. The higher incidence of lung cancer was the main cause of these findings. Moreover, drugs with both folic acid and vitamin B12 were observed to contribute to an increased mortality rate from all causes. The increased cancer mortality as well as the statistically insignificantly higher noncancer mortality were the primary causes of the second observation

Line 722: efficiency = efficiency.

We modified it as you recommended.

Line 743: TBW (Total Body Weight) or IBW (Ideal Body Weight)

We made the changes according to your suggestions.

Line 774: p.o. = What does that mean?

It is an excellent question. It is a veterinary drugs formula administration p.o. - per os (by mouth).

Line 794: perspectives  = perspectives.

Thank you. We have made the change suggested by you.

Line 1031: You must correct the bibliography (n. 89). The DOI is part of the previous article.

Thank you very much. Your observation helped us a lot. We appreciate it. We modified.